# Exon–Intron Differential Analysis Reveals the Role of Competing Endogenous RNAs in Post-Transcriptional Regulation of Translation

**DOI:** 10.3390/ncrna7020026

**Published:** 2021-04-16

**Authors:** Nicolas Munz, Luciano Cascione, Luca Parmigiani, Chiara Tarantelli, Andrea Rinaldi, Natasa Cmiljanovic, Vladimir Cmiljanovic, Rosalba Giugno, Francesco Bertoni, Sara Napoli

**Affiliations:** 1Institute of Oncology Research, Faculty of Biomedical Sciences, Universita`Svizzera Italiana, 6500 Bellinzona, Switzerland; nicolas.munz@ior.usi.ch (N.M.); luciano.cascione@ior.usi.ch (L.C.); chiara.tarantelli@ior.usi.ch (C.T.); andrea.rinaldi@ior.usi.ch (A.R.); francesco.bertoni@ior.usi.ch (F.B.); 2SIB Swiss Institute of Bioinformatics, 1015 Lausanne, Switzerland; 3Computer Science Department, University of Verona, 37129 Verona, Italy; luca.parmigiani@studenti.univr.it (L.P.); rosalba.giugno@univr.it (R.G.); 4PIQUR Therapeutics AG, 4057 Basel, Switzerland; natasa.cmiljanovic@swissrockets.com (N.C.); vladimir.cmiljanovic@swissrockets.com (V.C.); 5Oncology Institute of Southern Switzerland, 6500 Bellinzona, Switzerland

**Keywords:** miRNAs, lncRNAs, ceRNAs, translation, post-transcriptional regulation, mTOR pathway

## Abstract

Stressful conditions induce the cell to save energy and activate a rescue program modulated by mammalian target of rapamycin (mTOR). Along with transcriptional and translational regulation, the cell relies also on post-transcriptional modulation to quickly adapt the translation of essential proteins. MicroRNAs play an important role in the regulation of protein translation, and their availability is tightly regulated by RNA competing mechanisms often mediated by long noncoding RNAs (lncRNAs). In our paper, we simulated the response to growth adverse condition by bimiralisib, a dual PI3K/mTOR inhibitor, in diffuse large B cell lymphoma cell lines, and we studied post-transcriptional regulation by the differential analysis of exonic and intronic RNA expression. In particular, we observed the upregulation of a lncRNA, lncTNK2-2:1, which correlated with the stabilization of transcripts involved in the regulation of translation and DNA damage after bimiralisib treatment. We identified miR-21-3p as miRNA likely sponged by lncTNK2-2:1, with consequent stabilization of the mRNA of p53, which is a master regulator of cell growth in response to DNA damage.

## 1. Introduction

The transcriptome is tightly regulated at different levels: along with the regulation of new transcription, RNA molecules can be modulated at the post-transcriptional level, and noncoding RNAs play a relevant role in this process [1,2]. The most famous mechanism of post-transcriptional regulation is mediated by microRNAs, which are small noncoding RNAs that inhibit translation of mRNAs or induce their degradation [3]. In addition, long noncoding RNAs (lncRNAs) take part in this mechanism. Some of them possess miRNA responsive elements (MREs) [4] and are known as competing endogenous RNAs (ceRNAs). Indeed, they can act as an endogenous miRNA decoy, and their expression modulates the amount of free miRNAs available to bind their targets [5]. Circular RNAs (circRNAs) constitute a subclass of exceptionally stable ceRNA molecules that contain a covalent circular structure formed by noncanonical 3′ to 5′ end-joining event called back-splicing. circRNAs are diffuse and sometimes conserved in eukaryotic organisms [6]. High throughput analysis of protein expression, as liquid chromatography mass spectrometry, detects changes in translation efficiency, but it is expensive and requires specific expertise. Here, we show that also transcriptome profiling can help us to identify post-transcriptional modulations. RNA-Seq can be used to investigate pre-mRNA dynamics, taking advantage of the intronic sequences that are also acquired although less abundantly than exonic sequences. Changes in the overall intronic read counts directly measure changes in transcriptional activity. We can discriminate whether RNA levels are modulated at the transcriptional and post-transcriptional level applying an exon–intron split analysis (EISA), which compares intronic and exonic changes across different experimental conditions [7]. Here, we applied this approach to determine whether post-transcriptional regulation mediated by lncRNAs might be an additional layer to quickly control protein expression in diffuse large B cell lymphoma (DLBCL) cells exposed to bimiralisib, a dual PI3K/mTOR inhibitor with proven preclinical and early clinical anti-lymphoma activity [8,9]. The mTOR pathway regulates cell growth and proliferation in response to mitogen, nutrient, and energy status and therefore controls the balance between anabolism and catabolism in response to environmental conditions [10,11,12]. In addition to various anabolic processes as protein, lipid and nucleotide synthesis, mTOR also promotes cell growth by suppressing protein catabolism. mTOR is a downstream mediator of several growth factor and mitogen-dependent signaling pathways but also reacts to intracellular stresses that are incompatible with growth such as low ATP levels, hypoxia, or DNA damage.

The exposure of DLBCL cell lines to bimiralisib induces important post-translational and transcriptional changes, affecting genes and proteins involved in the PI3K/AKT/mTOR pathway, BCR signaling, NF-kB pathway, mRNA processing, apoptosis, cell cycle, Myc pathway, MAPK/RAS signaling, and glycolysis [8]. Most of the pathways are largely modulated via both transcription regulation and protein phosphorylation changes [8]. We hypothesized that post-transcriptional modulation can play a role in the mechanism of action of bimiralisib also. A fast adaptation to lack of nutrients requires an optimization of the stability of transcripts that need to be translated into proteins. Here, we describe how bimiralisib induced a quick stabilization of transcripts needed to cope with amino acid deficiency and to modulate the translation. This event strongly correlated with the overexpression of a lncRNA, namely lncTNK2-2:1, which is associated to the increased stability of transcripts affected by mTOR inhibition and responsible for DNA damage response. In particular, we validated the stabilization of p53 transcript due to the sponge effect on miR21-3p mediated by lnc-TNK2-2:1.

## 2. Results

### 2.1. Bimiralisib Reduces the Transcription of Genes Coding for Proteasome and Ribosome Components

We have previously reported that dual PIK3/mTOR pharmacological inhibition using bimiralisib has in vitro and in vivo anti-tumor activity and that it induces transcriptional changes in DLBCL cell lines [8]. Here, we have performed total RNA-Seq on RNA extracted from two DLBCL cell lines, U2932 and TMD8, exposed to DMSO or to bimiralisib for 4, 8, or 12 h. We noticed a general reduction for the most part of transcripts encoding for subunits of RNA pol I and III, which are responsible of rRNA transcription and of some subunits of RNA pol II, and of many transcripts encoding for the machinery processing rRNA. PIK3/mTOR inhibition induced also a downregulation of proteasome components, which was probably to balance the reduction of protein synthesis due to impaired ribogenesis (Appendix A).

### 2.2. Post-Transcriptional Regulation of Many Transcripts Encoding for Riboproteins and Translation Regulators Is an Early Event after Dual PIK3/mTOR Inhibition

We applied EISA [7] to transcriptomic changes upon bimiralisib treatment. EISA measures changes in mature RNA and pre-mRNA reads across different conditions to quantify the transcriptional and post-transcriptional regulation of gene expression. After 4 h of PIK3/mTOR inhibition, we observed a marked post-transcriptional regulation, since many transcripts showed independent changes in exons and introns. After 8 h, changes in transcript levels between DMSO or bimiralisib-treated samples have been mainly due to alteration of transcription, as evident by the fact that changes in exons were well correlated with change in introns (Figure 1a). Focusing on transcripts early stabilized after PIK3/mTOR inhibition, we found they were mainly encoding ribosome components involved in the response to amino acids starvation. Among the most quickly degraded transcripts, we found mRNAs encoding for spliceosome components and pre-mRNA processing (Figure 1b and Appendix A).

### 2.3. The lincRNAs RP11-480A16.1 (lncTNK2-2:1) and GMDS-AS1 Are Differentially Expressed after Dual PIK3/mTOR Inhibition and Strongly Correlated to Significantly Stabilized Transcripts

We hypothesized that a prompt alteration in the stability of several transcripts could be achieved by the expression of lncRNAs acting as miRNA sponges. We found 20 significant lncRNAs differentially expressed after pharmacological PIK3/mTOR inhibition: 15 upregulated and five downregulated. We also evaluated whether the 20 lncRNAs could be circRNAs, applying the algorithm CiriQuant [13] that accurately determines the junction of circRNAs from RNA-Seq paired samples. The output showed 28,521 back-spliced junctions (BSJ), most of which were cell specific (TMD8: 9519, U2932: 14,348). Six of the differentially expressed lincRNAs were among the reliably quantified circRNAs (Figure 2a, Table 1). We calculated the Pearson correlation index of each selected lincRNAs with each stabilized or degraded transcript (Δex/Δintr ≠ 1). The expression of two of these lncRNAs, namely RP11-480A16.1 (lncTNK2-2:1) and GMDS-AS1, strongly correlated with transcripts differentially stabilized upon bimiralisib exposure (Figure 2b and Appendix A). The post-transcriptionally modified transcripts, ranked by their correlation index with lncTNK2-2:1 and GMDS-AS1, were enriched in genes involved in the regulation of translation and in the response to amino acid starvation (Table 2). In particular, there was a highly significant enrichment of genes affected by mTOR inhibitor rapamycin, highlighting the prominent role of lncTNK2-2:1 and GMDS-AS1 in the post-transcriptional regulation due to bimiralisib exposure. We also noticed the enrichment of genes involved in DNA damage, and in particular, some of them were regulated by miRNAs as well. We focused on p53 and ATM due to their important biologic roles (Figure 3 and Appendix A).

### 2.4. lncTNK2-2:1 Induces Stabilization of p53 and ATM by Sequestering miR21-3p

We validated by qRT-PCR the correlation observed between the expression of lncTNK2-2:1 and GMDS-AS1 and the stability of ATM and p53 mRNAs. We measured the levels of the pre-mRNAs or the mature transcripts of ATM and p53 genes and the expression of lncTNK2-2:1 and GMDS-AS1, in U2932 and TMD8, at 4 h and 8 h after exposure to 1 µM of bimiralisib or DMSO. We confirmed the upregulation of both GMDS-AS1 and lncTNK2-2:1, although the GMDS-AS1 upregulation was significant in U2932 but not in TMD8 (Figure 4a). We measured the fold change of lncRNAs and of pre mRNAs or mature transcripts (Appendix A) and calculated their Pearson correlation index (Appendix A). We clearly confirmed the predicted correlation of the expression of the lncRNA lncTNK2-2:1 with the mature transcripts for p53 and ATM (R = 0.581 and R = 0.596, respectively) (Figure 4b, bottom panels) but not with their pre-mRNAs (Figure 4b, top panels). GMDS-AS1 was significantly correlated neither to p53 nor to ATM stability (Figure 4b). According to this, we searched for experimentally validated miRNA binding sites sheared by lncTNK2-2:1 and p53 in DIANA tools, LncBase (https://diana.e-ce.uth.gr/lncbasev3 accessed 15 April 2021) [4] and TarBase (https://carolina.imis.athena-innovation.gr/diana_tools/web/index.php?r=tarbasev8%2Findex accessed 15 April 2021) [14] (Appendix A). miR-21-3p and miR-22-3p are reported to target both lnc-TNK2-2:1 and p53 (Figure 4c), and miR-21-3p belongs to the miR-21-3p/TSC2/mTOR regulatory axis [15]. Thus, we focused on miR-21-3p, which was consistently reduced in both U2932 and TMD8 after bimiralisib treatment (Figure 4d) and the consequent upregulation of lncTNK2-2:1 (Figure 4a). The p53 stabilization was more robust in TMD8 than in U2932 (Appendix A), even if the lncRNA was upregulated in both cell lines (Figure 4a), which was an observation that was possibly justified by the lower basal expression of miR21-3p in U2932 compared to TMD8 (Appendix A).

### 2.5. lncTNK2-2:1 Degradation Reverts Stabilization of p53 and Releases miR21-3p

To validate the relationship between lncTNK2-2:1 expression and miR21-3p activity on its target p53, we electroporated 2 million TMD8 with 100 pmol of antisense oligonucleotides or of the negative control. After 72 h, we exposed the cells to 1 µM of bimiralisib or DMSO for 8 h. We measured the levels of the pre-mRNA or the mature transcript of the p53 gene and the expression of lncTNK2-2:1 by qRT-PCR. We confirmed that the antisense oligonucleotides degraded lncTNK2-2:1 efficiently both at basal condition and after its induction due to bimiralisib treatment (Figure 4e, left). We validated the relationship between the lncRNA lncTNK2-2:1 with the p53 transcript stability. Indeed, after lncTNK2:2-1 interference, we noticed the reduction of p53 stabilization (Figure 4e, middle), which was measured as the fold change between p53 mature mRNA with respect to the total pre-mRNA transcribed in each condition. According to this, we measured miR-21-3p after lncTNK2-2:1 degradation, and we showed that it increased in samples where lncTNK2-2:1 was knocked down with respect to the negative control (Figure 4e, right). This experimental observation confirmed the predicted miR-21-3p/lncTNK2-2:1/p53 regulatory axis.

## 3. Discussion

As a central controller of cell growth, mTOR regulates ribosome biogenesis. The latter is the most energy-demanding cellular process, and mTOR controls it by promoting the translation of riboproteins and by affecting ribosomal RNA (rRNA) synthesis. Ribosome synthesis requires all three nuclear RNA polymerases, Pol I for the synthesis of rRNA, Pol II for transcription of riboprotein genes, and Pol III for the synthesis of 5S RNA [16,17]. PIK3/mTOR inhibition by bimiralisib leads to the downregulation of all of them, a reduction of rRNA gene transcription, and, in addition, pre-rRNA processing impairment. Moreover, we describe here an additional mechanism that the cell activates when it must save energy: mTOR pathway inhibition leads to the stabilization of already present transcripts encoding for riboproteins. Since the new transcription of riboproteins genes is inhibited, the cell needs to save the already available riboprotein mRNA as long as possible in order to still translate essential proteins.

A growing number of miRNAs have been shown to control components or regulators of ribosome biogenesis [18]. In addition, lncRNAs have been increasingly found to play relevant roles in eukaryotic ribosome biogenesis that can be basally active or stress response-specific [18]. These molecular mechanisms include protein binding, rDNA chromatin modifications, snoRNP formation, and transcript-specific translation modulations. Here, we report evidence of an additional example of lncRNAs involved in the regulation of ribogenesis. Two lncRNAs, lncTNK2-2:1 and GMDS-AS1, were modulated following PIK3/mTOR pathway inhibition, and their transcription changes are strongly correlated with the stabilization of transcripts encoding for many riboproteins. Other lncRNAs were significantly upregulated by bimiralisib treatment but were not correlated with transcripts early stabilized after mTOR inhibition. Thus, we postulated that these particular lncRNAs could be relevant players of mTOR-mediated modulation of translation in response to amino acid deficiency or other stressful events. One of the possible mechanisms that might mediate a quick transcript stabilization is the sequestration of miRNAs, and GMDS-AS1 is already known as an miRNA sponge in lung cancer [19]. As proof of principle, we looked for binding sites in the lincRNAs for miRNAs that could target stabilized transcripts correlated with the upregulation of lncTNK2-2:1 or GMDS-AS1. Both lncTNK2-2:1 and GMDS-AS1 were associated with the stabilization of genes involved in DNA damage response and regulated by miRNAs. Among them, p53 is also involved in ribosome biogenesis [20,21], and the p53 network is known to interact with several miRNAs [22]. We could validate the correlation between p53 mRNA stabilization and lncTNK2-2:1 level but not with GMDS-AS1.

We showed that miR-21-3p increased after lncTNK2-2:1 silencing, and concomitantly, p53 was not stabilized after bimiralisib treatment in the absence of lncTNK2-2:1. This evidence enforced our hypothesis that miR-21-3p was responsible for p53 mRNA degradation, which we formulated on the basis of in silico base pairing of lncTNK2-2:1 and miR-21-3p and on the high stability of predicted miR-21-3p binding in p53 3′UTR, which is compatible with mRNA degradation, instead of inhibition of translation [23]. Furthermore, in U2932, a cell line expressing low miR-21-3p levels, p53 was not strongly stabilized, despite the upregulation of lncTNK2-2:1. In support of our findings, miR21-3p is already known to modulate the mTOR pathway via TSC2 mRNA downregulation [15], and P53-dependent mTOR inhibition is mediated by TSC2 [24]. Here, we provide the evidence of an additional layer of regulation of p53-mTOR crosstalk through the rapid elimination of miR-21-3p and consequent stabilization of p53 and enhancement of TSC2 repressor activity of mTOR. The PI3K/mTOR pharmacological inhibition enforces the mTOR inhibition by a positive feedback loop mediated by the overexpression of lncTNK2-2:1.

We also selected ATM as potential interesting transcript, since it appeared related to lncTNK2-2:1 overexpression both in silico and in vitro, but we could not identify any miRNA that may mediate the connection between the lncRNAs and the mRNA. ATM mRNA stabilization after bimiralisib exposure might also be due to the downregulation of miRNAs directly regulated by PI3K/signaling mTOR and directly targeting ATM 3′UTR [25].

In conclusion, based on an alternative bioinformatic approach applied to RNA-Seq data, we selected candidate molecules that could be involved in a post-transcriptional mechanism of RNA competition, and we provided data suggesting a novel RNA network composed by lncRNAs, miRNAs, and mRNAs, which is affected by the dual PI3K/mTOR pharmacological inhibition in DLBCL cell lines.

## 4. Materials and Methods

### 4.1. Cell Culture and Bimiralisib Treatment

Established human DLBCL cell lines TMD8 and U2932 were grown as previously described [8]. Bimiralisib was kindly provided by PQR Therapeutics (Basel, Switzerland). TMD8 and U2932 were seeded 3 million cells/well in a non-tissue culture 6 well plate. Cells were treated for 4 and 8 h with 1 µM bimiralisib or 0.1% DMSO (Sigma, St. Louis, MO, USA), respectively. Treatment was stopped by washing the cell with RNAse-free PBS and followed by immediate RNA extraction.

### 4.2. lncTNK2:2-1 Degradation

Three different locked nucleic acid (LNA) antisense oligonucleotides were designed against lncTNK2:2-1 and purchased by IDT (Integrated DNA Technology, Coralville, IA, USA) as 3-10-3 Affinity Plus (locked nucleic acid) gapmer format (3 Affinity Plus bases, 10 DNA bases, 3 Affinity Plus bases, phosporotioate bonds), along with a negative control. In details, their sequences were lncTNK2-2:1_ASO-1: CACTTCCCGAGTATAA; lncTNK2-2:1_ASO-2: CACCTGACCATATTGA; lncTNK2-2:1_ASO-3: CACCACTACACGTTTA; NC5 3-10-3: GACTATACGCGCAATA. TMD8 were nucleofected with 100 pmol of each antisense oligonucleotides or the negative control using 4D Nucleofector (Amaxa-Lonza, Basel, Switzerland), according to the manufacturer’s instructions and incubated for 72 h. Then, cells were treated with 1 µM of bimiralisib (PQR Therapeutics, Basel, Switzerland ), or DMSO (Sigma, St. Louis, MO, USA) for 8 h, and then, RNA was extracted.

### 4.3. RNA-Extraction

For each cell line and condition, cells were collected and resuspended in 1 mL of TRI Reagent (Sigma, St. Louis, MO, USA) for cell lysis, and extraction was performed. DNAse (Qiagen, Hilden, Germany) was added to the RNA samples and incubated for 15 min at room temperature. Total RNA was reprecipitated to remove salts and the enzyme.

### 4.4. Whole-Transcriptome Sequencing (RNA-Seq)

Two cell lines, U2932 and TMD8, were treated with 1 µM of bimiralisib or DMSO and RNA was extracted after 4, 8, or 12 h. Cells treatment and RNA extraction were described in [8]. Quality control for extract RNA was performed on the Agilent BioAnalyzer (Agilent Technologies, Santa Clara, CA, USA) using the RNA 6000 Nano kit (Agilent Technologies, Santa Clara, CA, USA), and concentration was determined by the Invitrogen Qubit (Thermo Fisher Scientific, Waltham, MA, USA) using the RNA BR reagents (Thermo Fisher Scientific). The TruSeq RNA Sample Prep Kit v2 for Illumina (Illumina, San Diego, CA, USA) was used for cDNA synthesis and the addition of barcode sequences. The sequencing of the libraries was performed via a paired end run on a NextSeq500 Illumina sequencer (Illumina). As an average, 25 million reads were collected per each sample.

### 4.5. Data Mining

We evaluated the RNA-seq reads quality with FastQC (v0.11.5), and we removed low-quality reads/bases and adaptor sequences using Trimmomatic (v0.35). The trimmed-high-quality sequencing reads were aligned using STAR [26], which is a spliced read aligner that allows for sequencing reads to span multiple exons. On average, we were able to align 85% of the sequencing reads for each sample to the reference genome (HG38). Then, the HTSeq-count software package [27] was used for the quantification of gene level expression. Differential expression analysis was performed on gene-level read count data using the ‘limma’ pipeline [28,29] We first subsetted the data to genes that had a counts-per-million value greater than one in 3 or more samples. The data were normalized per sample using the ‘TMM’ method from the edgeR package [30] and transformed to log2 counts-per-million using the edgeR function ‘cpm’. Then, linear model analyses, with empirical-Bayes moderated estimates of standard error, were used to identify genes whose expression was most associated with phenotype of interest, and an FDR-adjusted *p*-value of <0.05 was set as a threshold for statistical significance.

Transcription rates were estimated based on the number of nascent unspliced transcripts using EISA [7]. For each gene, we used HT-Seq to compute the average number of reads mapping to the gene’s introns (all exonic regions are excluded). Then, this number of intronic reads was divided by the total length of introns to yield a mean intronic coverage that was used as a proxy of the transcription rate.

Functional annotation was performed using Gene Set Enrichment Analysis (GSEA) [31] with all genes preranked by FC as determined by Limma test, or by delta exon/delta intron ratio as determined by EISA, or by Pearson correlation index between delta exon/delta intron ratio and lincRNAs expression. Gene sets were considered significantly enriched if *p* < 0.05 and FDR < 0.25.

The Pearson correlation was used to determine the relationship between exons/introns and gene expression. All statistical analysis was done with R (version 4.0.3) scripts. The significance of gene set overlap was determined by hypergeometric test.

MicroRNA binding prediction was performed searching in TarBase or LncBase and then intersection of miRNAs were computed by Venn diagram. MiRNA responsive elements (MRE) were calculated by the algorithm RNA22.

Prediction and quantification of circRNAs was carried out by the means of CIRIquant [13] with default parameters. In brief, CIRIquant uses HISAT2 [32] to align the RNA-seq reads to the reference genome and CIRI2 to identify putative circRNAs in the form of BSJ, which are then filtered to reduce the number of false-positives. Since the normalization of circRNA expression values is necessary for the differential expression analysis, TMM normalization factors were extracted from gene expression levels to remove the systematic technical effect of library size. Gene count matrix for the normalization was obtained using the script prepDE.py, from stringTie [33], on the aligned reads. Finally, the voomWithQualityWeights [34] function, from the limma package in Bioconductor [28], is applied to identify the statistical significance of circRNA expression change.

### 4.6. Reverse Transcription of Total RNA to cDNA

Total RNA (500 ng) was processed for each sample by the SuperScript III First-Strand Synthesis SuperMix for qRT-PCR (Invitrogen, Carlsbad, CA, USA) according to the manufacturer’s instructions. cDNA was stored at −20 °C.

### 4.7. Quantitative Real-Time PCR

qRT-PCR was performed on an Applied Biosystem StepOnePlus System. LncRNA and mRNA targets expression were quantified using KAPA SYBR FAST qPCR Master Mix (2×) ABI Prism 5 mL (KAPA Biosystems, Wilmington, MA, USA) according to the manufacturer’s instructions, and the comparative CT method (∆∆CT method) normalized to ACTB (β-Actin) expression was applied for data analysis. The following primers were used: GMDS-AS1, forward: 5′-CCC AGT CTT CCC AGG ATT GA-3′, reverse: 5′-AGC ATC TTC CAG GCC AAA TG-3′; lncTNK2-2, forward: 5′-AGA GCG AAA CCC CAT CTC AA-3′, reverse: 5′-GGA GAA GGA AGC GGA CTG AT-3′; ACTB, forward: 5′-CCA ACC GCG AGA AGA TGA C-3′, reverse: 5′-TGG GGT GTT GAA GGT CTC A-3′; ATM, pre-mRNA forward: 5′-AAC CAC AGT TCT TTT CCC GT-3′, pre-mRNA reverse: 5′-TTG ACT CTG CAG CCA ACA TG-3′, mRNA forward: 5′-GCC TTA AAA CTT TGC TTG AGG TG-3′, mRNA reverse: 5′-ACA TGC GAA CTT GGT GAT GA-3′; TP53, pre-mRNA forward: 5′-ACA AGC AGT CAC AGC ACA TG-3′, pre-mRNA reverse: 5′-AGA GCA ATC AGT GAG GAA TCA G-3′, mRNA forward: 5′-ACA AGC AGT CAC AGC ACA TG-3′, mRNA reverse: 5′-CAC CAC CAC ACT ATG TCG AAA A-3′. miR-21-3p and U6 snRNA expression were quantified using TaqMan microRNA Assay (Applied Biosystems, Foster City, CA, USA) and TaqMan microRNA Control Assay, respectively, according to the manufacturer’s instructions, and the comparative CT method (∆∆CT method) normalized to U6 expression was applied for data analysis. PCR efficacy was determined using the LinRegPCR tool [35].

## Figures and Tables

**Figure 1 ncrna-07-00026-f001:**
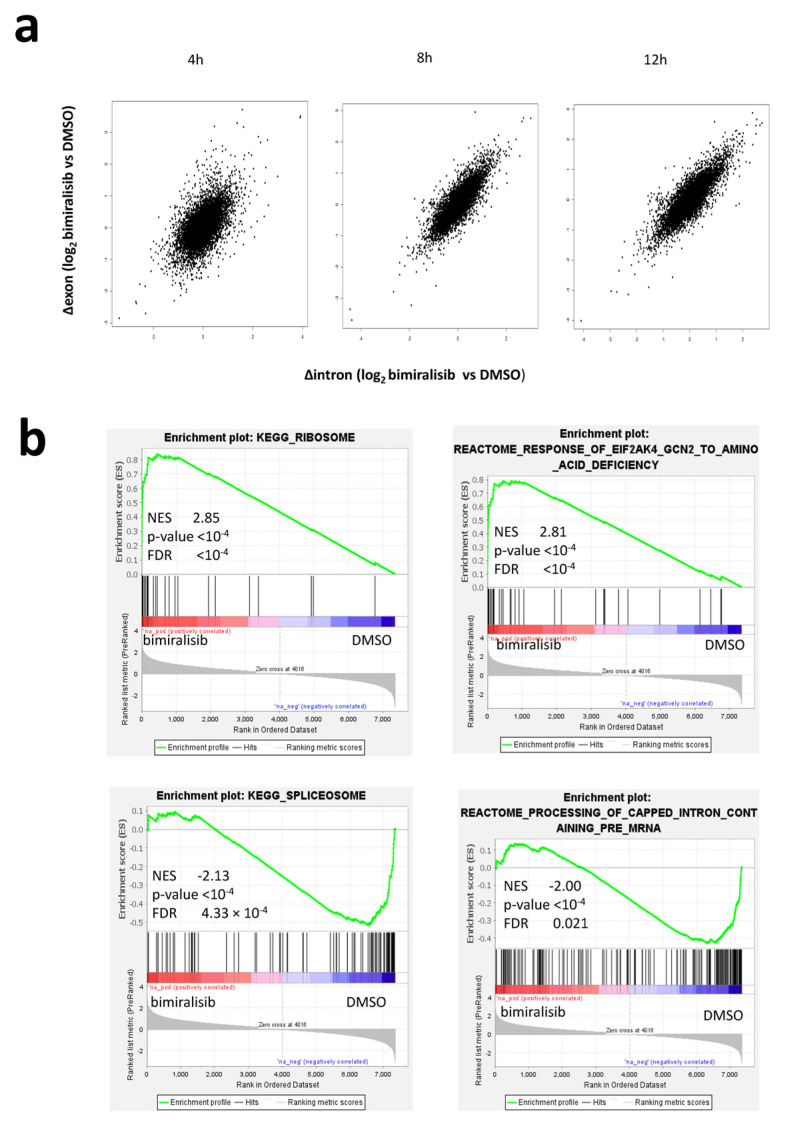
Post-transcriptional regulation of transcripts encoding for riboproteins (RPs) and translation regulators is an early event after dual PI3K/mTOR inhibition. (**a**) Scatter plots representing the comparison of changes in intronic (Δintron) and exonic (Δexon) reads per each expressed transcript in DLBCL cell lines exposed to bimiralisib or DMSO for 4, 8, and 12 h. Δexon/Δintron = 1 reflects transcriptional modulation, Δexon/Δintron ≠ 1 reflects post-transcriptional modulation. (**b**) Representative GSEA plots illustrating the transcriptional expression signature enrichment in transcripts ranked by their decreasing Δexon/Δintron ratio in DLBCL cell lines exposed to bimiralisib (1 µM) or DMSO for 4 h. Green line, enrichment score; bars in the middle portion of the plots show where the members of the gene set appear in the ranked list of genes. Positive or negative ranking metrics indicate correlation or inverse correlation with the profile, respectively. FDR, false discovery rate; NES, normalized enrichment score.

**Figure 2 ncrna-07-00026-f002:**
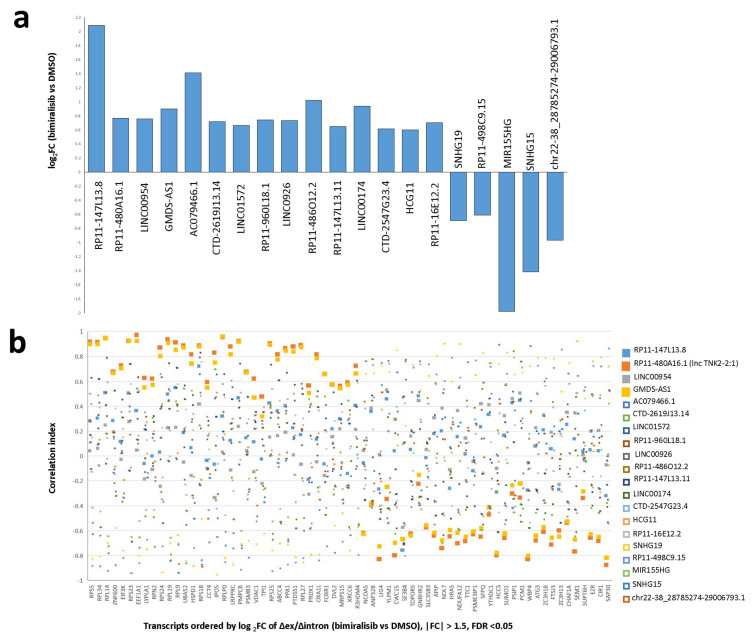
RP11-480A16.1 (lncTNK2-2:1) and GMDS-AS1 are lincRNAs differentially expressed in bimiralisib vs. DMSO strongly correlated to the stabilization of same transcripts. (**a**) Fold change of expression levels of lincRNAs differentially expressed at any time point in U2932 and TMD8 treated with bimiralisib (1 µM) or DMSO. (**b**) Plot of Pearson correlation indexes calculated for Δexon/Δintron ratio of significant differentially stabilized transcripts and levels of differentially expressed lincRNAs. Orange and yellow filled squares represent correlation indexes referred to lncTNK2-2:1 and GMDS-AS1, respectively. Blue and gray-filled squares represent correlation indexes referred to RP11-147L13.8 and LINC00954, respectively. Empty squares refer to all the other significant differentially expressed lincRNAs. LncTNK2-2:1 and GMDS-AS1 are lincRNAs differentially expressed in bimiralisib vs. DMSO strongly correlated to the stabilization of same transcripts.

**Figure 3 ncrna-07-00026-f003:**
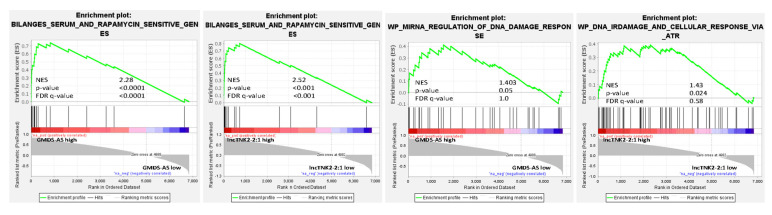
Stabilized transcripts in correlation to lncTNK2-2:1 and GMDS-AS1 expression are involved in the regulation of translation and DNA damage. Representative GSEA plots illustrating the transcriptional expression signature enrichment in transcripts stabilized in correlation to lncTNK2-2:1 and GMDS-AS1 high expression. Green line, enrichment score; bars in the middle portion of the plots show where the members of the gene set appear in the ranked list of genes. Positive or negative ranking metrics indicate the correlation or inverse correlation with the profile, respectively. FDR, false discovery rate; NES, normalized enrichment score.

**Figure 4 ncrna-07-00026-f004:**
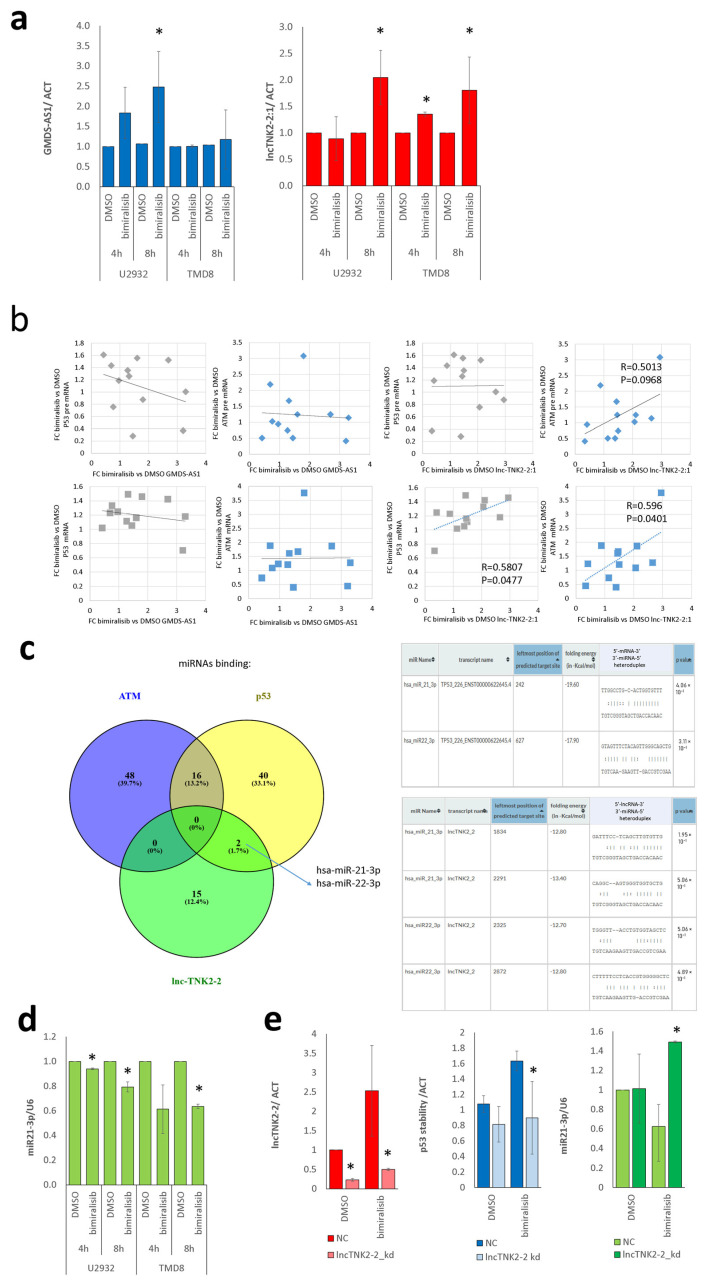
lncTNK2-2 induces the stabilization of p53 by sequestering miR21-3p. (**a**) Expression levels of GMDS-AS1 (top) and lncTNK2-2:1 (bottom) in U2932 and TMD8 after 4 h and 8 h of exposure to DMSO or bimiralisib (1 µM). (**b**) Pearson correlation of fold change of GMDS-AS1 or lncTNK2-2:1 expression levels (x axis) after bimiralisib treatment with fold change of pre-mRNA (top) or mature mRNA (bottom) of ATM or p53 (y axis). Correlation index (R) and *p* value (P) are indicated in the plot when significant. (**c**) Left, Venn diagram of common miRNAs binding p53, ATM, and lncTNK2-2:1. Right, RNA22 prediction of miRNA binding sites of miR21-3p or miRNA 22-3p in p53 3′UTR or lncTNK2-2. (**d**) Expression level of miR21-3p in U2932 or TMD8 after 4 h or 8 h of exposure to DMSO or bimiralisib (1 µM). (**e**) Expression levels of lncTNK2-2:1 (left), p53 stability (middle), and miR21-3p (right) after 8 h of bimiralisib exposure in negative control or lncTNK2-2:1 knockdown cells. * *p* < 0.05.

**Table 1 ncrna-07-00026-t001:** Table summarizing the statistical analysis of differentially expressed lincRNAs and their circRNA reconstruction.

Gene_Name	Ensembl ID	logFC	AveExpr	t	*p* Value	adj. *p* Value	circRNA
RP11-147L13.8	ENSG00000267731.1	2.08	4.1	9.74	7.00 × 10^−7^	6.48 × 10^−5^	NO
RP11-480A16.1	ENSG00000260261.1	0.763	4.62	9.52	8.89 × 10^−7^	7.32 × 10^−5^	YES
LINC00954	ENSG00000228784.6	0.76	3.5	7.74	7.03 × 10^−6^	0.000205749	YES
GMDS-AS1	ENSG00000250903.7	0.9	3.75	7.27	1.30 × 10^−5^	0.000290741	YES
AC079466.1	ENSG00000266976.1	1.41	4.25	7.06	1.71 × 10^−5^	0.000341307	NO
CTD-2619J13.14	ENSG00000232098.3	0.72	4.42	6.4	4.24 × 10^−5^	0.000611259	NO
LINC01572	ENSG00000261008.5	0.66	3.77	6.35	4.54 × 10^−5^	0.000636107	YES
RP11-960L18.1	ENSG00000261218.4	0.74	3.5	6.29	4.93 × 10^−5^	0.000670722	YES
LINC00926	ENSG00000247982.5	0.73	4.98	5.82	9.95 × 10^−5^	0.001049071	YES
RP11-486O12.2	ENSG00000247373.3	1.02	3.53	5.8	0.0001	0.001063727	NO
RP11-147L13.11	ENSG00000278730.1	0.65	4.7	5.5	0.00016	0.001445039	NO
LINC00174	ENSG00000179406.6	0.94	3.52	5.29	0.00022	0.001840434	NO
CTD-2547G23.4	ENSG00000274925.1	0.62	3.55	4.57	0.00071	0.004099891	NO
HCG11	ENSG00000228223.2	0.6	3.59	4.48	0.00084	0.004600193	NO
RP11-16E12.2	ENSG00000259772.5	0.7	3.93	4.3	0.00115	0.005710491	NO
SNHG19	ENSG00000260260.1	−0.68	4.14	−5.4	0.00018	0.001630628	NO
RP11-498C9.15	ENSG00000263731.1	−0.61	4.96	−5.59	0.00014	0.001328078	NO
MIR155HG	ENSG00000234883.3	−1.97	5.66	−9.15	1.33 × 10^−6^	8.69 × 10^−5^	NO
SNHG15	ENSG00000232956.7	−1.41	5.37	−9.89	5.98 × 10^−7^	6.00 × 10^−5^	NO
chr22-38_28785274-29006793.1	ENSG00000279978.1	−0.97	5.8	−10.21	4.28 × 10^−7^	5.64 × 10^−5^	NO

**Table 2 ncrna-07-00026-t002:** Table of top gene sets enriched after preranked gene set enrichment analysis (GSEA) of transcripts ordered by Pearson correlation index between Δexon/Δintron ratio and expression levels of lncTNK2-2:1 or GMDS-AS1.

Name	Size	ES	NES	NOM *p*-Value	FDR *q*-Value	FWER *p*-Value
REACTOME_EUKARYOTIC_TRANSLATION_ELONGATION	37.000	0.721	2.522	0.000	0.000	0.000
REACTOME_RESPONSE_OF_EIF2AK4_GCN2_TO_AMINO_ACID_DEFICIENCY	42.000	0.682	2.419	0.000	0.000	0.000
REACTOME_SELENOAMINO_ACID_METABOLISM	50.000	0.633	2.304	0.000	0.000	0.000
REACTOME_EUKARYOTIC_TRANSLATION_INITIATION	51.000	0.610	2.219	0.000	0.000	0.000
REACTOME_ACTIVATION_OF_THE_MRNA_UPON_BINDING_OF_THE_CAP_BINDING_COMPLEX_AND_EIFS_AND_SUBSEQUENT_BINDING_TO_43S	24.000	0.681	2.205	0.000	0.000	0.000
REACTOME_NONSENSE_MEDIATED_DECAY_NMD_	52.000	0.593	2.187	0.000	0.000	0.000
REACTOME_SRP_DEPENDENT_COTRANSLATIONAL_PROTEIN_TARGETING_TO_MEMBRANE	54.000	0.577	2.133	0.000	0.000	0.002
REACTOME_FANCONI_ANEMIA_PATHWAY	26.000	0.645	2.112	0.000	0.000	0.002
REACTOME_INFLUENZA_INFECTION	74.000	0.499	1.932	0.000	0.004	0.032
REACTOME_HDR_THROUGH_SINGLE_STRAND_ANNEALING_SSA_	23.000	0.580	1.873	0.000	0.011	0.109
REACTOME_ASSOCIATION_OF_TRIC_CCT_WITH_TARGET_PROTEINS_DURING_BIOSYNTHESIS	28.000	0.540	1.791	0.000	0.034	0.308
REACTOME_HDR_THROUGH_HOMOLOGOUS_RECOMBINATION_HRR_	39.000	0.512	1.777	0.000	0.037	0.358

## Data Availability

Profiling data are available at the National Center for Biotechnology Information (NCBI) Gene Expression Omnibus (GEO) (http://www.ncbi.nlm.nih.gov/geo accessed 15 April 2021) database.

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
