# Peer review of "Exon–Intron Differential Analysis Reveals the Role of Competing Endogenous RNAs in Post-Transcriptional Regulation of Translation"

_ncrna, 2021, doi:10.3390/ncrna7020026_

Round 1

Reviewer 1 Report

In their manuscript, Munz et al. aim to investigate the contribute of post-transcriptional regulation to the mechanisms of adaptation response induced by stressful conditions. The authors treated diffuse large B cell lymphoma cell lines with bimiralisib, a dual PI3K/mTOR inhibitor, to simulate growth stress conditions and explored post-transcriptional regulation by exon-intron differential analysis. The manuscript addresses an interesting question, shows interesting results and is well argumented and written. However some criticisms need to be solved before it can be considered suitable for publication in this journal.

This study started from the identification of a)genes coding for proteasome and ribosome components as classes of transcripts post-transcriptional regulated after dual PIK3/mTOR inhibition and b) 20 lncRNAs differentially modulated after bimiralisib treatment. Looking for a specific mechanism of post-transcriptional stabilization that is the ability of circular lncRNAs to act as ceRNAs, the authors focalized the study on one lncRNA, lncTNK2-2:1, which resulted associated to the increased stability of transcripts affected by mTOR inhibition and pointed to miR-21-3p as miRNA likely sponged by lncTNK2-2:1. To experimentally validate this regulatory axis the authors treated the two DLBCL cell lines with the PI3K/mTOR inhibitor, and quantified the expression level not only of the lncTNK2-2:1 and of miR21-3p, but also of p53 and ATM genes as representative of the post-transcriptionally modulated class of transcripts. Again they showed the expected correlation among the components of this regulatory circuit, but this is not a true validation of the interactions of the components of this circuit. To support this axis the authors say: (lanes 216-217 of the discussion) “We believe that miR-21-3p was likely responsible of p53 mRNA degradation, since in U2932, a cell line lowly expressing miR-21-3p, p53 was not strongly stabilized, despite the upregulation of lncTNK2-2:1”. The modulation of axis components  and the analysis of the consequences of this modulation is indeed the way for experimentally validating this regulatory circuit. The authors should, after bimiralisib cell treatment, block the consequent lncTNK2-2:1 increase using a specific siRNA and verify the modulation of miR21-3p and transcriptional and post-transcriptional regulation of the miRNA targets. Vice versa, the expression level of miRNA and genes should be verified in DLBCL cells overexpressing lncTNK2-2:1  .

Minor points  Legends of the figures are often too synthetic. Moreover the legend should contain the spelling of all the acronyms used in the figures.

Author Response

In their manuscript, Munz et al. aim to investigate the contribute of post-transcriptional regulation to the mechanisms of adaptation response induced by stressful conditions. The authors treated diffuse large B cell lymphoma cell lines with bimiralisib, a dual PI3K/mTOR inhibitor, to simulate growth stress conditions and explored post-transcriptional regulation by exon-intron differential analysis. The manuscript addresses an interesting question, shows interesting results and is well argumented and written. However some criticisms need to be solved before it can be considered suitable for publication in this journal.

This study started from the identification of a)genes coding for proteasome and ribosome components as classes of transcripts post-transcriptional regulated after dual PIK3/mTOR inhibition and b) 20 lncRNAs differentially modulated after bimiralisib treatment. Looking for a specific mechanism of post-transcriptional stabilization that is the ability of circular lncRNAs to act as ceRNAs, the authors focalized the study on one lncRNA, lncTNK2-2:1, which resulted associated to the increased stability of transcripts affected by mTOR inhibition and pointed to miR-21-3p as miRNA likely sponged by lncTNK2-2:1. To experimentally validate this regulatory axis the authors treated the two DLBCL cell lines with the PI3K/mTOR inhibitor, and quantified the expression level not only of the lncTNK2-2:1 and of miR21-3p, but also of p53 and ATM genes as representative of the post-transcriptionally modulated class of transcripts. Again they showed the expected correlation among the components of this regulatory circuit, but this is not a true validation of the interactions of the components of this circuit. To support this axis the authors say: (lanes 216-217 of the discussion) “We believe that miR-21-3p was likely responsible of p53 mRNA degradation, since in U2932, a cell line lowly expressing miR-21-3p, p53 was not strongly stabilized, despite the upregulation of lncTNK2-2:1”. The modulation of axis components  and the analysis of the consequences of this modulation is indeed the way for experimentally validating this regulatory circuit. The authors should, after bimiralisib cell treatment, block the consequent lncTNK2-2:1 increase using a specific siRNA and verify the modulation of miR21-3p and transcriptional and post-transcriptional regulation of the miRNA targets. Vice versa, the expression level of miRNA and genes should be verified in DLBCL cells overexpressing lncTNK2-2:1

AU . We thank the Reviewer for the positive comments and we agree that the manuscript had to be improved. We performed the suggested experiment to validate the modulation of axis components. We inhibited the upregulation of lncTNK2-2:1 by three different LNA antisense oligonucleotides and we observed both at basal level and after bimiralisib treatment the reduction of p53 stabilization and the increase of miR-21-3p. Unfortunately, due to the lack of time and of an already available plasmid for lncTNK2:2-1 overexpression, we could not perform the inverse experiment.

Minor points  Legends of the figures are often too synthetic. Moreover the legend should contain the spelling of all the acronyms used in the figures.

AU . We apologize for the mistake. We have modified the legends.

Reviewer 2 Report

            The submitted manuscript by Cascione et al shows indirect evidence for a role of lincRNA lncTNK2-2:1 in sequestering (sponging) miR21-3p with up regulation of lncTNK2-2:1 and mRNA stabilization after bimiralisib 20 treatment. This is a paper to add to the growing body of published experiments showing a three-way regulation or cross-talk involved with lncRNAs, miRNAs and mRNAs. However the paper would have been strengthened if it showed experimental verification of the interaction of lncTNK2-2:1 with miR21-3p.

Suggestions/corrections to improve the paper:

1.Title: perhaps change to "Exon-intron differential analysis is consistent with the role……."

   2. The manuscript is very hard to read. To help the reader, many terms used should be better defined, e.g., in Abstract, p. 1, line 13 define mTOR, mechanistic target of rapamycin (mTOR) and explain it better in the Introduction for those outside the field; line 18 explain "exon-intron differential analysis."; line 38, Proteomics is a broad term, define exactly what you mean; p. 8 line 186 define RP (ribosomal protein?), why not just spell it out as there is an overabundance of abbreviations used that makes reading the text difficult.   p. 2, line 70 "we validated the stabilization of p53 transcript due to the sponge effect on 70 miR21-3p mediated by lnc-TNK2-2:1." validate? May be an overstatement. Perhaps use: "propose the stabilization of p53 transcript due to the predicted sponge effect" as there are no experiment data to support the sponge effect, e.g., the use of a luciferase reporter assay. Also, any experiments done to show lnc-TNK2-2:1 is found localized in the cytoplasm, which would be consistent with a sponge effect?     

3.  Figure 1. I recommend better describing the techniques used to obtain the data. Add explanations in the caption. Figure 1b drawings are hard to understand.      

4. I would move Fig.S2 b, which shows the predicted lncRNA/miRNA interactions, to the main text to provide more evidence related to the sponging proposal and and have it right upfront. Should not the bases be in RNA and not DNA notation? Also, Fig.S2b should show which RNA chains in the heteroduplex diagrams are lncRNA and miRNA. A long shot, but are there any phylogenetic data available to show mutated miRNA or lncRNA in one or more of the base-pairings and with base-pair compensatory changes, e.g. G:U --> U:A or to G:C? This would strength the conclusions.     

5. Table 2     chr22-38_28785274-29006793.1 is confusing, and it is indeed an unusual way Ensembl uses to describe this RNA gene.  Maybe call  the gene ENSG00000279978 and ID transcript  ENST00000623644, this way the reader could easily find the gene as one would not think that a gene would have this name, which actually represents coordinates in chr22. Also, Ensembl lists no transcript for ENSG00000228784.6 but four transcripts for LINC00954 but with other names, see http://may2015.archive.ensembl.org/Homo_sapiens/Gene/Summary?db=core;g=ENSG00000228784;r=2:19868860-19885047     

6. Fig 3. 

It is difficult to read Fig. 3. Please describe better in the caption what the two bottom drawings are that are on top of each other, how to read them. I could see a correlation with the top diagram but an explanation of the results (and showing relationship to p53, which is not mentioned in the diagram) would help. Best to add the reference and/or website for the program(s) used to obtain the data in the figure caption.

7. Add website address for search programs such as LncBase, p. 10 line 287. Although mentioned in the Introduction, it would be good to site the Paraskevopoulou et al reference again here.

8. The spelling seems mostly OK but recheck, e.g., p 9, line 233 bionformatic

Reviewer 3 Report

This manuscript describes the bioinformatic analysis of RNA-seq data from two diffuse large B cell lymphoma cell lines treated for different times with a dual PI3/mTOR inhibitor.  From their analysis of exon versus intron reads using the EISA protocol they draw a number of provocative conclusions about post-transcriptional regulation and the role of a circular competing endogenous RNA in mRNA stabilization via miRNA titration.  While much bioinformatic data is presented, I am not at all convinced by the conclusions and a number of significant weaknesses were noted.  Most importantly, the analysis pipeline ignores an important confounding issue, some of the most important conclusions involve very modest changes in data values, and interpretative models are not supported by any independent data. The paper addresses many issues (transcriptional regulation, post-transcriptional regulation, translation, mRNA stability, lncRNA expression and miRNA function.  However, for each there is only a minimum of supporting data presented and RNA-seq analysis alone is not sufficient.  

Specific Comments.

  1. The title is awkward and suggests a general new mode of gene regulation that is not supported by the data.  Even if the data supported it, a better title might be :”Exon-intron differential analysis reveals the role of competing endogenous RNAs in post-transcriptional regulation of translation”.  If the story is a general one, it needs to be extended to more than just these two cell lines. 
  2. Most of the important data relies on the previously published EISA method of analysis, which compares exon-only ad intron-only reads to indicate post-transcriptional regulation of gene expression after drug treatment.  I have a serious concern about this sort of interpretation in the current project.  Drug treatment leads to many changes in gene expression, and at multiple levels owing to the inhibition of both PI2K and the mTOR pathway.  This is highly stressful on the cells. Further, the analysis of intron reads is complicated by a phenomenon not mentioned or considered by the authors - intron retention via splicing defects.  A paper in 2017 (Lee et al., Cell 171: 1545) reported that mTOR inhibition leads to splicing changes and importantly to widespread increased intron retention of many of the same genes and pathways discussed in this manuscript.  RNAs with retained introns are mostly kept in the nucleus. Interpreting intron reads as representing increased transcription may be incorrect as these reads may merely reflect altered splicing.  Intron retention is being more and more recognized as a key process in gene regulation.  There are many papers about this.  
  3. The experimental details are not clear.  How many replicates were were sequenced and at what depth?  The authors refer to a previous paper (reference 8) about this, but that paper also does not mention this sort of data.  Without it, the statistical analyses are impossible to interpret.  
  4. Their analysis identifies a number of noncoding RNAs showing differential expression, though most of these changes are less than twofold.  They select two circular RNAs for further study as their expression change correlates with that of mRNAs of interest.  They further hypothesize that these might alter the stability of mRNAs by acting as ceRNAs to “sponge” miRNAs.  One of the two, lncTNK2-2.1, may harbor binding site(s) (how many??) for miR21-3p but the other does not appear to be consistent with their model.  How is the expression of lncTNK2-2.1 correlated with the expression of its host gene?  If it’s a circRNA, does it have an intron, or only joined exons?  While the authors aim to present a model for lncRNA-mediated gene regulation, there is no compelling support for this.  In fact, they admit it (“We believe that miR-21-3p was likely responsible for p53 mRNA degradation…”). Such a model requires more.  
  5. Minor: there are a number of places where grammar needs improvement.

Author Response

Reviewer 3

This manuscript describes the bioinformatic analysis of RNA-seq data from two diffuse large B cell lymphoma cell lines treated for different times with a dual PI3/mTOR inhibitor.  From their analysis of exon versus intron reads using the EISA protocol they draw a number of provocative conclusions about post-transcriptional regulation and the role of a circular competing endogenous RNA in mRNA stabilization via miRNA titration.  While much bioinformatic data is presented, I am not at all convinced by the conclusions and a number of significant weaknesses were noted.  Most importantly, the analysis pipeline ignores an important confounding issue, some of the most important conclusions involve very modest changes in data values, and interpretative models are not supported by any independent data. The paper addresses many issues (transcriptional regulation, post-transcriptional regulation, translation, mRNA stability, lncRNA expression and miRNA function.  However, for each there is only a minimum of supporting data presented and RNA-seq analysis alone is not sufficient.  

Specific Comments.

  1. The title is awkward and suggests a general new mode of gene regulation that is not supported by the data.  Even if the data supported it, a better title might be :”Exon-intron differential analysis reveals the role of competing endogenous RNAs in post-transcriptional regulation of translation”.  If the story is a general one, it needs to be extended to more than just these two cell lines. 

AU: We performed the experiments to prove the relationship between lncTNK2-2:1 expression, miR21-3p level and p53 stabilization, and we believe that our theory is now more robust and corroborated by experimental data. Anyway, the aim of our paper is to provide an example of how uncovering relationships among different molecules, just analyzing already available data under different perspective. We describe here a mechanism of gene regulation which is already well described but we want to suggest an additional computational approach to identify more and more  interactions that are worth to be further investigated. In agreement with the Reviewer, we modified the title as suggested.

  1. Most of the important data relies on the previously published EISA method of analysis, which compares exon-only ad intron-only reads to indicate post-transcriptional regulation of gene expression after drug treatment.  I have a serious concern about this sort of interpretation in the current project.  Drug treatment leads to many changes in gene expression, and at multiple levels owing to the inhibition of both PI2K and the mTOR pathway.  This is highly stressful on the cells. Further, the analysis of intron reads is complicated by a phenomenon not mentioned or considered by the authors - intron retention via splicing defects.  A paper in 2017 (Lee et al., Cell 171: 1545) reported that mTOR inhibition leads to splicing changes and importantly to widespread increased intron retention of many of the same genes and pathways discussed in this manuscript.  RNAs with retained introns are mostly kept in the nucleus. Interpreting intron reads as representing increased transcription may be incorrect as these reads may merely reflect altered splicing.  Intron retention is being more and more recognized as a key process in gene regulation.  There are many papers about this.  

AU: We thank the Reviewer for the relevant comment. The EISA computational approach takes in account the average expression of all the introns of a certain transcript and in this way we can discriminate the intron retention events, that usually involve one or more introns but not all, from the measurement of the pre-mRNA. In ref 7 it is described as EISA approach was tested to assess its efficiency in nascent transcription estimation in three experiments in which GROseq was performed along with RNAseq.

Anyway we agree with the Reviewer that we must pay special attention in a dataset where splicing is affected, as in our case (Fig.1b). Extensive intron retention events are actually included in our estimation of destabilized transcripts (delta ex/delta intr <1). In our study we mostly focused on the correlation between the expression of lncRNAs and the increase of transcripts stability (delta ex/delta intr >1) due to miRNA sponge effect. It is true that an intron retention effect might have reduced the number of the predicted stabilized transcripts, but according to this, our analysis can be affected by false negative rather than false positive candidates. We are confident that our observations on stabilized transcripts are reliable.

  1. The experimental details are not clear.  How many replicates were sequenced and at what depth?  The authors refer to a previous paper (reference 8) about this, but that paper also does not mention this sort of data.  Without it, the statistical analyses are impossible to interpret.  

AU: We apologize for this. We included details in material and methods section.

  1. Their analysis identifies a number of noncoding RNAs showing differential expression, though most of these changes are less than twofold.  They select two circular RNAs for further study as their expression change correlates with that of mRNAs of interest.  They further hypothesize that these might alter the stability of mRNAs by acting as ceRNAs to “sponge” miRNAs.  One of the two, lncTNK2-2.1, may harbor binding site(s) (how many??) for miR21-3p but the other does not appear to be consistent with their model.  How is the expression of lncTNK2-2.1 correlated with the expression of its host gene?  If it’s a circRNA, does it have an intron, or only joined exons?  While the authors aim to present a model for lncRNA-mediated gene regulation, there is no compelling support for this.  In fact, they admit it (“We believe that miR-21-3p was likely responsible for p53 mRNA degradation…”). Such a model requires more.  

AU: In our work, we want to propose a pipeline to select molecules that can interplay in a regulatory circuitry, for further investigation. We applied the algorithm CiriQuant to our dataset to identify, among others, lncRNAs that could have an additional feature compatible with competing endogenous RNA behavior. We also interrogated available webtools, as LNCBase and RNA22, to identify putative interactions between selected lncRNAs and miRNAs. In particular for lncTNK2-2:1 we found two binding sites for miR-21-3p and two for miR-22-3p, as reported in Fig 4c. GMDS-AS1 was compatible with circRNA structure but it lacked evidence of correlation with the stabilization of p53 or ATM targets and of miRNA binding sites in common with the selected targets. For such reason we focused on lncTNK2:2-1 for further investigation. In this revised version of the manuscript, we provided an experimental evidence of lncTNK2-2:1 capability to affect the level of free miR-21-3p and consequent p53 stability. Even if interesting, it was not our priority, here, the experimental validation of lncTNK2-2:1 circularization. By CiriQuant we identified a back-splicing  event involving lncTNK2-2:1 along with other transcripts belonging to a transcriptionally complex region containing several pseudogenes, lncRNAs and antisense RNAs between TNK2 and TFRC genes in chr3: 195906294-196077122. Unfortunately, the algorithm failed to properly reconstruct the circRNA and we cannot provide more details about its structure, a part of the presence of a statistical significant back-splicing event in the comparison between DMSO and bimiralisib treated samples, involving lncTNK2-2:1

  1. Minor: there are a number of places where grammar needs improvement.

AU: We apologize for this. We revised grammar in the manuscript.

Round 2

Reviewer 3 Report

The authors have carried out additional experimental work and have generally responded well to my previous concerns.